# Flexible IoT Gas Sensor Node for Automated Life Science Environments Using Stationary and Mobile Robots

**DOI:** 10.3390/s21217347

**Published:** 2021-11-04

**Authors:** Sebastian Neubert, Thomas Roddelkopf, Mohammed Faeik Ruzaij Al-Okby, Steffen Junginger, Kerstin Thurow

**Affiliations:** 1Institute of Automation, University of Rostock, 18119 Rostock, Germany; thomas.roddelkopf@celisca.de (T.R.); steffen.junginger@uni-rostock.de (S.J.); 2Center of Life Science Automation (celisca), 18119 Rostock, Germany; mohammed.al_okby@atu.edu.iq (M.F.R.A.-O.); kerstin.thurow@celisca.de (K.T.); 3Technical Institute of Babylon, Al-Furat Al-Awsat Technical University (ATU), Kufa 51015, Iraq

**Keywords:** IoT, sensor node, life science, automation, laboratory, mobile robot, gas detection

## Abstract

In recent years the degree of automation in life science laboratories increased considerably by introducing stationary and mobile robots. This trend requires intensified considerations of the occupational safety for cooperating humans, since the robots operate with low volatile compounds that partially emit hazardous vapors, which especially do arise if accidents or leakages occur. For the fast detection of such or similar situations a modular IoT-sensor node was developed. The sensor node consists of four hardware layers, which can be configured individually regarding basic functionality and measured parameters for varying application focuses. In this paper the sensor node is equipped with two gas sensors (BME688, SGP30) for a continuous TVOC measurement. In investigations under controlled laboratory conditions the general sensors’ behavior regarding different VOCs and varying installation conditions are performed. In practical investigations the sensor node’s integration into simple laboratory applications using stationary and mobile robots is shown and examined. The investigation results show that the selected sensors are suitable for the early detection of solvent vapors in life science laboratories. The sensor response and thus the system’s applicability depends on the used compounds, the distance between sensor node and vapor source as well as the speed of the automation systems.

## 1. Introduction

Life science laboratories are still dominated by partial and island automation. The degree of automation can be increased by connecting different automation islands distributed in the laboratory building. The use of mobile robots will influence this development. Several applications using mobile robots in life science laboratories have been described [1,2,3]. The connection of originally separated automation plants which now needs to pass laboratories and corridors which in parallel are used by human operators leads to novel challenges regarding occupational safety. One main issue is the transfer of compounds, which partially emits hazardous or toxic gases. The labware for transporting such materials needs to be closed to avoid the compounds or resulting gases from escaping. However arising leakages, undetected contaminations or simple accidents, for example by collisions on the automation island (by local robots) or between mobile robots and obstacles, require a fast detection to avoid hazardous situations for the human operators.

A wide range of gases, which affect the health risk for humans belong to volatile organic compounds (VOC). This group of chemicals evaporates easily from fluids or solids at room temperature. VOCs can have natural or anthropogenic origins and occur for example in agriculture, in many industry branches, in traffic and also in common household products, for example, in paints, adhesives, cosmetics and cleaning agents [4,5]. Consequently, they are all around and can partially lead to serious health problems since a number of them exhibit toxic, neurotoxic, carcinogenic or mutagenic properties [6]. Furthermore, the concentration of many VOCs are consistently up to ten times higher indoors than outdoors [7] because of the often higher emission and missing air exchange. In life science industries/laboratories many VOCs, for example, C_6_H_14_ (hexane), CH_3_OH (methanol), C_2_H_5_OH (ethanol), CH_2_Cl_2_ (dichloromethane), C_3_H_6_O (acetone), C_2_H_3_N (acetonitrile), C_2_H_6_O (dimethyl ether), C_3_H_8_O (2-propanol), C_7_H_8_ (toluene), are widely used as the starting material, as solvents, as refrigerants or for dilutions and many further VOCs in partially high concentrations can be found there. Consequently, the risk for health impairments [8] in such environments can be significantly higher if they are not handled correctly. Thus, it is necessary to monitor the often autonomously acting systems, especially if the automation systems can leave the laboratories equipped with appropriately safety features.

Many research groups deal with the field of gas detection and monitoring. This high interest is on the one hand attributable to novel innovations and possibilities in sensor design [9,10] and on the other hand to the increasing demand of technical solutions, for example, the air pollution monitoring in urban environments [11,12,13], the supporting measures against the COVID-19 pandemic [14,15] and the protection of working environments against hazardous situations by the exposure of pollutants [16,17,18].

Benammar et al. presented a solution for gas monitoring in the private sector. Therefore, a modular IoT platform for the real-time monitoring of the indoor air quality regarding CO_2_ (carbon dioxide), CO (carbon monoxide), SO_2_ (sulfur dioxide), NO_2_ (nitrogen dioxide), O_3_ (trioxygen) and Cl_2_ (chlorine) was developed. The platform is based on the low-power *Waspmote* hardware (Libelium, Zaragoza, Spain) architecture in combination with a XBee *PRO* Series 2 radio module for data transmission. By using analog front-end (AFE) modules the gas sensors are arranged on a sensor interface board, which is connected to the Waspmote board. A gateway device based on the Raspberry Pi 2 B (Raspberry Pi Foundation, Cambridge, UK) minicomputer allows the adaption to Internet standards like WiFi and Ethernet for HTTP data transmission to an IoT server [19].

Addabbo et al. developed a low-power gas sensor node for detecting CO, O_2_ (oxygen) and NO_x_ (nitrogen oxides) in industrial plants and public buildings (including temperature and humidity). For the integration of the battery-driven sensor nodes a multi-layer network architecture on the basis of ZigBee (low-power near field protocol) and LoRa (Long Range; low-power wide-range protocol) communication was used. The ZigBee sensor nodes on the lower layer build mesh sub-networks and transfer data via gateways directly to the Internet or to the next higher layer, which uses LoRa. On this layer LoRa-based sensor nodes and ZigBee-LoRa gateways submit data via the LoRa-Internet gateway to the upper layer of the Internet backbone. The data were transferred via HTTP (hypertext transfer protocol) and stored in a MySQL-database; the functionality of the solution regarding data acquisition and transfer was tested by establishing a network in the institute’s building (7000 m^2^ with five floors) [20].

A further gas sensor node was presented by Zhao et al., which focused on monitoring the indoor air quality (IAQ) by detecting CO_2_, PM_2.5_ (particulate matter with a diameter of 2.5 μm or less [21]) and HCOH (formaldehyde). Concerning the data transmission a multi-protocol approach supporting Modbus (via RS485; wired communication) and LoRa, GPRS (general packet radio service), WiFi, and the NB-IoT (Narrowband IoT; low-power wide-area (LPWA) technology) was implemented. This allowed for the sensor-node integration via local networks or the Internet. The data were transferred into a cloud platform where they were stored in a database and published by a web server [22].

A further IoT sensor node (named iAQ+) was presented by Marques et al., which focused on the occupational health in laboratory environments regarding the air quality. The iAQ+ prototype sampled the IAQ index (rating system for indoor air quality) by using the smart sensor BME680 from Bosch (Gerlingen, Germany; including temperature, humidity and air pressure measurement). As a basis the WiFi-ready development board Fire Beetle ESP8266 was used. A web service collected, analyzed and stored the data in a SQL Server database and notified the user via SMS (short message service) or e-mail if the limit values were exceeded [23].

Wall et al. introduced a gas monitoring IoT solution for promoting wellness and safeguarding social interaction. In that development the BME680 from Bosch was also used to detect the IAQ index as seen in Marques et al. The data were transmitted to a server (hosted on a Raspberry Pi) via HTTP POST request (WiFi) and stored in a MySQL database. The solution was evaluated by two 2-week data collection periods in a kitchen and a study [24].

An electronic nose using four smart sensors was presented in Arroyo et al. The sensors IAQ-Core and CCS811 from AMS (Premstätten, Austria) as well as the SGP30 from Sensirion (Staefa, Switzerland) and the BME680 from Bosch were integrated for measuring VOC and peripheral parameters on one compact circuit board. The data were transferred via Bluetooth low energy to a mobile device where they were interpreted by a neural network integration [25]. A similar sensor combination was also used by Jose et al., using LoRa for data transmission [26].

The developments presented show several compact gas detecting IoT-solutions for different application areas. These solutions chiefly are not adaptive enough, for example, regarding the fast integration of new sensors or the adaption of other power supplies, as required for the gas detection and expectation of other environmental data acquisitions in the automation infrastructure of a life science laboratory. Due to that, in this paper a flexible IoT (Internet of Things) gas-sensor node, using a modular functional concept, is presented. In the pursued application scenario very critical situations with fast and serious consequences for laboratory assistants can arise, which consequently require a fast detection of small leakages wherever they occur. So the flexible integration of the sensor node into primary actors of the automation environment is required.

## 2. System Concept and Implementation

The basis of the considerations in this paper is a modular sensor node, which has a major demand to be easily adapted or extended to different measurement scenarios in life science automation. This includes, for example, the option for wired and wireless communication, the fast exchange of sensors and the adaption of different power supplies. One first measurement aim is the reliable and fast detection of VOC emissions in laboratory environments especially in cases where the sensors are in motion. As detection range for the sensors, the level 4 (>3–10 mg/m^3^—660—2200 ppb) and level 5 (>10–25 mg/m^3^—2200—5500 ppb) of the recommended guide values from the German Federal Environment Agency [27] and concentrations beyond that, which are commonly occur in laboratory environments, need to be covered by the target IoT-solution. However, a highly precise concentration determination is not required and the differentiation of the detected VOCs is not yet focused. Accordingly, a continuous TVOC-monitoring is used for the current target application. A further important demand is the continuous data transfer to the automation infrastructure for the immediate data interpretation. Small size and long battery life of the sensor node are secondary demands in this paper.

### 2.1. Gas Sensor Node

The developed sensor node consists of four modules with different functionalities. These modules can be individually combined to a stack depending on the specific requirements of the applications. The size of one module is 50 × 35 × 12 mm^3^ and the resulting stack reaches a height of 50 mm (excluding battery, see Section 2.1.4). In Figure 1 the components of the sensor node are briefly introduced.

#### 2.1.1. Microcontroller Board

The central module of the sensor node is the microcontroller board primary for data sampling and processing and for managing the data transfer and the system configuration via the USB-C-interface. It consists of an ARM Cortex M0+ microcontroller (32-bit, 32 MHz) including I^2^C—inter-integrated circuit, SPI—serial peripheral interface, and UART—universal asynchronous receiver transmitter. The USB interface is also used for the power supply if the battery board is not available. As the central unit of the node the microcontroller is connected to all pins for inter-board connection (see Section 2.1.5). The configuration via the USB interface allows the adjustment of the calibration settings for each implemented sensor or the configuration of the settings for WiFi-communication if any changes from the default settings are necessary. Further, this board has a real-time clock (MCP7941-I/MS, Microchip Technology, Chandler, AZ, USA) included to enable absolute time stamps for the acquired data and to avoid postponements during data sampling, especially during the beginning of the measurement.

#### 2.1.2. Sensor Board

The second required module is the sensor board, which includes the sensors and the required voltage adaption. The sensor board’s design follows a simple structure and can be designed individually with different sensors, depending on the application whereby it can be exchanged or extended (adding more than one sensor module into the node stack) if required. Currently the sensor board is equipped with three sensors as shown in Table 1 and has space still to be extended depending on the individual sensor’s size and peripheral requirements.

The selected metal-oxide semiconductor gas sensors (MOX), BME688 and SGP30, are tiny digital solutions which already handle the heater control, calibration procedures, baseline and long-term correction, humidity compensation (for BME688 partially supported by a related processing library [28]). They also offer comfortable interfaces as SPI or I^2^C. In investigations from Yurko et al. about BME680 and SGP30 it can be seen that the sensors show suitable characteristics for the pursued application regarding consistence and reproducibility [31]. The BME688 is an enhancement of the BME680, which includes all features of the BME680 as well as some additional features, for example, options for using artificial intelligence [32]. In addition to gas sensors, other sensors can also be integrated on the board to extend the sensor node’s monitoring range. This applies to temperature and humidity sensors, optical or acoustic sensors. One example is the high-resolution atmospheric pressure sensor MS5803-05BA (*TE connectivity*, Schaffhausen, Switzerland) for the detection of changings in the height of the sensor node’s position (required in combination with the mobile robot)), which is also used in the current configuration.

The sensor board exhibits a unique shape in the form of two extension arms for the gas sensors. This allows a better contact with the target gases and avoids a mutual temperature influence of the sensors due to their heating procedures. Both sensors are currently used with the lowest supported sampling rate of 1 Hz. The sampling rate can be increased, but then the data interpretation needs to be performed by the user themselves.

In contrast to the SGP30 the BME688 data interpretation is not part of the sensor itself, here, the already mentioned library is required, for example, to convert the measured IAQ (index of air quality)-value into a TVOC-concentration or to compensate the influence of ambient temperature and humidity, which affects the emission rate of VOCs [33]. Correspondingly, the ambient parameters of the BME688, also including atmospheric pressure, are provided to the library for the compensation of such influences. In the presented development the Bosch library bsec_2-0-6-1_generic_release_04302021 is used. The SGP30 also accepts the relative humidity as an input parameter for compensation purposes, which is taken in the current configuration from the BME688.

#### 2.1.3. Communication Board

The communication board is optional and allows, besides the USB communication (microcontroller board), the wireless communication for the sensor node. Currently the *ESP-WROOM-02D* WiFi-module (Espressif Systems, Shanghai, China) and the ACN52840 Bluetooth 5 module (Aconno, Düsseldorf, Germany) are integrated. The WiFi-module is used for data transmission while the BLE-module will be used for indoor localization matters, which will not be studied further in this paper.

The WiFi-module includes an own microcontroller which deals with the wireless communication to the infrastructure cloud. It is connected via SPI to the microcontroller board to set the real-time clock by the servers’ clock time, to request the configuration settings (for example, available sensors, WiFi connection data) and to transfer the acquired sensor data including the absolute time stamp. Further a FRAM (ferroelectric random access memory)-storage (512 kByte) is considered on this board for buffering data in case of an interrupted network connection.

#### 2.1.4. Power/Battery Board

A further optional module is the power/battery board. It offers, additionally to USB, two further options for the sensor node’s power supply. A direct supply of 24 V can be connected by a terminal block (two poles). This enables the connection with devices such as mobile robots, which do not support USB for the power supply of peripheral devices. The second option is the supply from a single cell battery (3.7 V, 53 × 32 × 9 mm^3^, currently with 1000 mAh), which can be recharged via the USB interface of the microcontroller board or via the 24 V if necessary.

#### 2.1.5. Inter-Board Connection

To support a higher flexibility regarding the integration and combination of the single boards, every board consists of two board-to-board connector lines on each side, which allow a firm attachment between them and permit access to the required pins to every board, independent of the boards’ stack order. In Figure 2 the pin-assignment for the inter-board connection is shown. Extending boards can simply be attached and integrated via the available interface solutions. If additional communication interfaces on one of the boards are required then this does not necessarily need to be connected to the inter-board connection. This is, for example, performed for the FRAM-storage integration on the communication-board since this is only required for the WiFi communication.

### 2.2. Communication

If the basic sensor node, including microcontroller and sensor board, is used in combination with the communication-board, the wireless data transfer via WiFi to the infrastructure cloud can be supported. In the sensor node the microcontroller board establishes the internal communication to the available sensors and waits for data requests from the communication board. The communication between these boards is realized by SPI in which the communication board takes on the master role. Before the communication board starts with the data transfer it inquires communication parameters, such as target network, IP-addresses and ports from the microcontroller board. Since the microcontroller board is the central module all important configuration parameters are stored on it. In case the communication board needs to be exchanged the sensor node keeps the individual configuration parameters. If the communication board has all required parameters, it listens for data requests from the communication server of the infrastructure cloud. The communication server sends a UDP broadcast in a configurable interval including server identification, current time stamp, requested data (e.g., all data including ambient temperature and humidity or gas data only) and the URL for HTTP-data transmission. The advantage of this communication strategy is that the sensor nodes do not need to be registered at the server. A sensor node can be started directly with the necessary network-configurations and all further information is provided via the broadcast. The supplied current time stamp also permits the synchronization of the measurement data if communication errors occur and data need to be buffered in the FRAM.

If the configuration procedure is finished, the repetitive data transfer is started as seen in the sequence diagram of Figure 3. The microcontroller board collects the data from the sensors and transfers them to the communication board, where all data are directly stored in the FRAM. If the communication board receives the UDP-broadcast the required data are selected and bundled in a structured JSON-protocol which includes base information and a list of the data sets (see Figure 4). Every data set consists of the requested data for one measurement period (currently 1 s), separated for the individual sensors. Appropriately, for every data set one time stamp is considered. For every single value the name and the unit is supplied additionally.

On the server side the received data packets are assigned and stored into the relational database structure of the infrastructure cloud. One advantage of using higher structured protocols is that new, unknown sensor parameters or sensor nodes can be automatically included without administration support. The data are available via the provided IoT-Web-App including a visualizing interface that allows viewing certain periods of the acquired data or a live view of incoming data (see Figure 5).

## 3. Experimental Methods and Results

The aim of the experiments in this paper is to prove the usability of the selected sensors and to define application conditions for using the developed sensor node solution in an automated environment. Therefore, investigations under laboratory conditions and application-related experiments were executed in a classical laboratory environment. For all experiments the sensor node was given a 30 min warm-up time to achieve stable sensitivity level of MOX-sensors [34].

### 3.1. Investigations under Laboratory Conditions

The investigations under laboratory conditions were executed inside a Secuflow fume hood (Waldner Holding GmbH & Co. KG, Wangen im Allgäu, Germany). The fume hood was required to avoid the distribution of the arising fumes and to avoid strong influences by circulations of the ambient air. During the experiments the Secuflow was closed and the exhaust ventilation was temporarily turned off. In the experimental design the sensor node was attached to a stand of corresponding height above a petri dish (see Figure 6a). The dispensing of the VOCs was performed by different pipettes from Eppendorf (Hamburg, Germany) depending on the respectively required amount. The laboratory was equipped with an air conditioning system which kept the temperature at 22.0 ± 0.5 °C and the relative humidity at 50.0 ± 2.0%.

Both sensors have different baselines in neutral environments. The SGP30 fluctuates between 0–0.05 ppm in the fume hood whereas the BME688 baseline is between 0.49–0.51 ppm whereby the minimal value of the sensor is given with 0.5 ppm by the Bosch-library.

#### 3.1.1. Sensor Orientation

A first key issue is to identify the preferred orientation of the sensor node in relation to the emission source. Starting from the fact that in our applications the emission is primarily located below the sensor node, the assumption is that in this case a higher detection sensitivity can be reached if the sensor is faced downwards. In the experiment setting, the sensors were fixed at a height of 25 cm above the petri dish. The sensors’ orientations (facing upwards, sideways and downwards; see Figure 6) were investigated with two VOCs, ethanol and hexane, which had varying effects on the sensors regarding the sensitivity. For every experiment an amount of 1 mL (for hexane) and 10 µL (for ethanol) were dispensed into the petri dish and positioned directly below the sensors. The chosen amounts were defined in preliminary investigations, so that the sensors show clear results but do not reach their saturation.

In Figure 7 the results of the orientation experiment are presented. From the data it can be seen that the facing-downwards orientation always shows a strong response compared to the facing-upwards orientation, as to be expected. The facing-sideways orientation shows partially stronger reactions than the downwards orientations, but this behavior is not reliable since in some cases its reaction can be weaker than the face upwards orientation, as it can be seen in Figure 7 for the BME688 sensor and hexane.

#### 3.1.2. Reactivity of Sensors

In a second consideration the reactivity of the used sensors regarding arising VOC emissions are compared. For this experiment the same setting was used as described in Section 3.1.1. The height of the sensors was 25 cm above the petri dish and the sensor was facing downwards. In Figure 8 the sensors’ reactivity is exemplarily shown for hexane, as VOC with a weaker response, and ethanol, as VOC with a stronger response to the sensors.

The presented data show that both sensors react almost parallel to hexane. The amplitudes differ only insignificantly. For ethanol the SGP30 shows a fast rising and decay behavior in contrast to the BME688. For abruptly increasing concentrations the BME688 shows a strong deceleration for increase and decrease, whereby the decrease case down to the baseline may take several minutes, depending on the amplitude.

#### 3.1.3. Sensors’ Reaction Related to Different VOCs

In this consideration the sensors’ behavior on five different VOCs depending on the distance between sensor and petri dish (height) and the amount of the compounds were investigated. The environmental conditions here were the same as in the investigations before. The sensor node was faced downwards oriented and the petri dish was directly underneath. The following commonly used solvents and compounds were tested:C_2_H_5_OH—ethanol (70%, technical grade)CH_2_O_2_—formic acid (≥98%)CH_2_Cl_2_—dichloromethaneC_2_H_3_N—acetonitrileC_6_H_14_—hexane

In the investigation setting two different heights, 25 and 40 cm, and four different amounts for each compound were used. For dichloromethane, acetonitrile and hexane the amounts 1, 5, 10 and 20 mL were selected. In comparison to the other compounds, ethanol and formic acid showed significantly stronger responses to the sensors in the investigation setting. Very low amounts of these compounds led very fast to the saturation of the sensors, especially for the BME688. Due to this behavior, the test amounts for ethanol and formic acid were equally reduced to 10, 100, 500 and 1000 µL. In Figure 9, Figure 10, Figure 11, Figure 12 and Figure 13 the sensors’ responses to different compounds were shown, separated for both sensors and both heights. For higher concentrations the responses for ethanol and formic acid still reached the sensors saturation and were correspondingly not shown in Figure 9 and Figure 10.

For ethanol and formic acid both sensors showed, as expected, a strong sensitivity by reaching the maximum values (saturation) of the sensors (BME688: 1000 ppm; SGP30: 60 ppm). For lower amounts it can either be observed that the measured concentration also decreases or that the value decreases earlier from the maximum, since the smaller amounts of the compounds were already evaporated. Further, the results show that the range of the measured amounts, which fall within the scope between baseline and saturation, is mostly less than 100 µL, depending on the sensor. The influence regarding the different heights here can only be observed for formic acid where a small change of height strongly effects the measured concentration, which proves the sensors sensitivity.

Despite higher amounts for dichloromethane, hexane and acetonitrile, significantly smaller responses can be measured. For most measurements both sensors show a similar behavior. A clear exception is acetonitrile, here the BME688 responds first with a decreasing measurement value, which after differently long periods slowly increases. In contrast to that the SGP30 shows values that correlate with the amount of the compound. Regarding the different distances between the sensors and the petri dish only hexane shows a stronger response for 25 cm than for 40 cm.

### 3.2. Application-Related Investigations

These investigations were executed in an automated laboratory environment. In small application scenarios using one stationary and one mobile robot, the behavior of the sensor node was investigated. Additional to the laboratory investigations the gas exposure time onto the sensor node has an important influence, depending on the robot’s speed. Moreover, the environments are not free of other VOC influences, which partially effects the baselines of the sensors.

#### 3.2.1. Stationary Transport Robot: TS60 (Stäubli)

In this experiment the sensor node’s behavior in an application-related scenario using the SCARA TS60 from Stäubli was investigated. The sensor node was mounted to one side of the robots concentric three-finger gripper (see Figure 14a). In that simple pick-and-place application the robot arm moved from a distant position (distance: 94 cm) to a rack, which included one 15 mL tube. The arm had to grab the tube and to bring it back to the start position. The experiment was performed at four different speed levels, which can be set in fixed percentage steps (used steps: 2%, 5%, 10% and 25%) from the programmed maximum speed. SCARAs inherently offer a high-speed level depending on the mechanical structure of the joints, which is why the speed did not exceed 25%. For this task the TS60 primarily required two joints, the main rotatory joint (joint 1) and the translatory joint (joint 3) with the gripper on its bottom end (see Figure 14b). The nominal speeds of the axes were 385°/s for joint 1 and 2000 mm/s for joint 3. Consequently in the different speed levels the tasks require the following mean durations for execution (in min:sec:msec):2% speed: 1:16:525% speed: 00:31:1510% speed: 00:15:9325% speed: 00:06:84
Figure 14Experimental setting using the SCARA TS60. (**a**) Sensor node attached to the gripper which is above the tube. The petri dish is positioned opposite to the sensor node. (**b**) Joint 1 and 2 are rotatory joints and act around the *z*-axis. Joint 3 is a translatory joint and is acting in z-direction (up and down). Further the distance between the start position (also the end position) and the position of the tube to be picked up is shown.
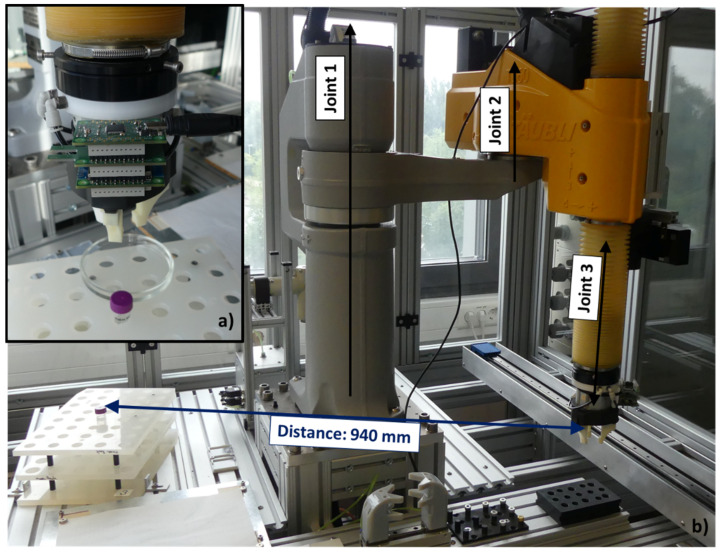


In this experiment again hexane and ethanol were used as VOCs, where hexane has a rather low and ethanol a rather high response to the integrated sensors. A petri dish (diameter: 10 cm) was positioned close to the tube on the opposite side, where the sensor node is mounted to the arm, when it is above the tube (see Figure 14a). In this way the worst-case scenario for detecting emissions around the target point was selected. In each experiment 1 mL hexane and 100 µL ethanol were used. Since the environment was not as encapsulated as in the fume hood, the baselines’ ranges partially dilated and reached 0–0.05 ppm for the SGP30 and 0.47–0.53 ppm for the BME688.

The data for 1 mL hexane show a different behavior between BME688 and SGP30 (see Figure 15). While the SGP30 has comparably slight responses for all speed levels, the BME688 clearly detects hexane for the speed levels 2% and 5% but not for the higher speed levels. The responses in these levels are too close to the baseline to use them for detection and the same applies to the SGP30 results. A reason for the failed detection is that the BME688 requires considerably more time for such VOCs with a weaker effect. Further, due to the fast movement an airflow arises, which in first order influences the measurement for the slower reacting sensor, BME688.

In contrast to that, 100 µL ethanol could be clearly detected by both sensors for all speed levels (see Figure 16). Also here the dependency of the BME688 from the speed level becomes apparent by the strong decreasing concentration with a higher speed. The SGP30 shows a distinctly more stable behavior regarding the amplitude over the speed levels and also a faster decay behavior.

#### 3.2.2. Mobile Robot: H20 (Dr. Robot)

In the last experiment the mobile robot H20 from Dr Robot (Ontario, ON, Canada) was used (see Figure 17a) to prove the sensors’ behavior by passing the VOC emission sources on the laboratory floor. For that the sensor node was once mounted to the side and once mounted to the front of the robot with each 25 cm above the floor. In case that the sensor node was side-mounted it directly passes above the VOC source while in case of the front-mounted sensor node additionally a lateral shift of around 40 cm arose (see Figure 17b,c). The investigation was executed with the three speed levels 0.14, 0.21 and 0.28 m/s, whereby 0.28 m/s corresponds with the standard transfer speed typically used.

As investigation compounds again, hexane and ethanol were used, which were provided in a petri dish (diameter: 15 cm). While 5 mL hexane was dosed into the petri dish, only 1 mL ethanol was used. Using smaller amounts in this experiment were unsuitable due to the fast evaporation of the compounds. The petri dish was placed a short distance from the H20 path so that the robot just passed it, as seen in Figure 17. For the investigation execution a laboratory was used, which did not include technical equipment with ventilators since these can influence the baseline. The baseline in the investigation laboratory was for the BME688 in the range of 0.45–0.55 ppm and for the SGP30 between 0–0.07 ppm.

In Figure 18 and Figure 19 the results of the VOC-concentration measurements for the prescribed speed levels are presented. The results are separated for the selected sensors and the method of mounting. In all scenarios a clear response from the sensors can be observed. As is to be expected, the results from the side-mounting for both sensors and for both compounds show stronger responses than from the front-mounting. For ethanol in combination with the side-mounting both sensors reach their saturation for all speed levels. Thus, only one speed level is presented in Figure 19a,b. The different speed levels only show insignificant differences or ambiguous responses between the sensors. This may depend on the relatively low difference of the speed levels and additionally it is assumed that the plain front surface of the robot’s travel unit (lower robot part) pushes the air and causes swirling effects, which influence the results. This effect can be clearly observed in Figure 19b where the SGP30 regarding its faster reactivity compared to the BME688 (Figure 19a) shows this behavior.

Since this experiment was executed in a real laboratory air drafts by the air conditioning or opening doors could have also slightly influenced the experimental results. Especially the presence of humans could influence the measurement significantly, because of the perspirations of cosmetics and the human body itself. Thus, these disruptive factors were avoided as much as possible.

## 4. Discussion and Conclusions

The developed IoT sensor node follows a modular approach, which can be adapted to various application scenarios regarding measured parameters and the functional features. Currently the sensor node consists of four modules from which two modules (microcontroller and sensor board) are necessary for the base functionality (sensor data acquisition, data processing, system configuration etc.) and two further modules (communication and battery board) are available to provide additional features, such as wireless communication and battery operation. Unnecessary modules can simply be excluded and the required ones can be attached to any position due to the multi-interface connection between the modules. Thus, a flexible IoT-platform is provided, which can be simply adapted and extended. In that way the sensor node can be used independently from the sensor configuration as a local sensor-node (connected to a PC), as an IoT-sensor node and as a mobile IoT-sensor node. The disadvantage of this modular approach is that the sensor node is comparably large and that by the divided processing units, a higher power consumption and consequently a lower battery life arises.

The data transfer to the infrastructure cloud is realized in a JSON format via HTTP, which is always initiated by a UDP-broadcast from the communication server. The strategy using a broadcast allows the server to control the data transfer (dividing to different target systems, adjustment of transfer cycle and transferred data) and to synchronize all connected sensor nodes. The JSON protocol is structured in a way that an automatic data assignment on the server side can be performed. Consequently, the protocol is comparably comprehensive, but for any system extensions regarding new parameters or new sensor nodes no manual adaptions on the server side are required.

The presented configuration of the sensor node focus the continuous early detection of hazardous gas situations by two integrated MOX gas sensors, which allow a TVOC-monitoring. To improve the fast detection of critical situations in machine-dominated areas the integration of such sensor nodes into the mechanic actors of automated environments were investigated.

In the executed investigations the suitability of the sensor node for detecting critical gas emissions in automated laboratory environments was examined. First of all the general characteristic of the selected gas sensors (SGP30 and BME688) was ascertained in laboratory investigations. This comprises the preferred sensors’ orientation if the approximate direction of the emission is known (here primarily below), the reactivity of the sensors and their behavior towards different frequently used VOCs in celisca (ethanol, formic acid, acetonitrile, dichloromethane and hexane). Regarding the orientation both sensors clearly show a stronger response if they are facing downward, into the direction of emission. In combination with a robot this orientation may possibly hold some dangers for the sensors if, for example, very high concentrations affect the sensors or if accidents induced by the robots pollute the sensors with any substances. In all cases a suitable casing is recommended.

Concerning the reactivity of the sensors the BME688 shows, compared to the SGP30, for high concentration differences a significantly higher delay, which also can be observed in the decay range. The reason for the delay is the interposed library and the therein included filtering. If the IAQ as a result is sufficient and stable ambient conditions are given, the Bosch library is quite possibly not mandatorily necessary and a faster response could be observed.

The investigations regarding the different VOCs have shown that the sensors react remarkably parallel, neglecting the delay of the BME688. One exception is acetonitrile where here the results obviously differ. The BME688 responds with a negative amplitude before it turns to slight positive amplitudes with a significant delay. As expected, between the VOCs very different responses could be noticed, whereby in the presented investigation ethanol and formic acid show significantly higher amplitudes than the other compounds.

In the practice-related investigation the sensor-nodes integration into the laboratory equipment was evaluated. Therefore, two kinds of robots were used, the SCARA TS60 (Stäubli) and the mobile robot H20 (Dr Robot). Both robots moved the sensor node and in the case of the TS60 partially very high speeds are reached. These high speeds show a clear influence on the gas detection, especially for compounds with rather low response to the sensors. A reason for this behavior is the considerable shorter time near the VOC emission source and the comparably slow sampling rate of the sensors (1 s). In combination with the delay of the BME688 in the fastest trials, hexane could not be detected. The results of the SGP30 certainly show corresponding results, but the amplitude only minimally exceeds the thresholds of the baseline. For 1 mL hexane a clear detection is only possible using lower movement speeds of the TS60. In case of 100 µL ethanol the detection can be clearly observed whereby the measured concentration strongly decreases with the increasing speed level. For all trials the fast movements of the TS60 gripper inducts an air draft and turbulences, which partially can be seen in the data by delayed or short, stronger sensor responses. In the case of fast moving robots it can be necessary to consider specific short-stop positions in the robots work flow, especially in critical areas, where the robot is passing very fast without waiting phases. Furthermore, regular robot waiting phases can be used to let the robot patrol in the safe area for unexpected emissions.

By using mobile robots the higher speed levels play a subordinated role since these robot types usually work cooperating in the same area as human operators. In contrast to stationary robots, which are fixed on tables and are partially encapsulated by a housing, the mobile type passes many different areas, which have no constant baseline and which are stronger influenced by environmental effects, such as humans or arising air drafts. In this investigation the robots passing an emission source with different speeds and the influence of two mounting positions was tested. It could be observed that especially for the side-mounting, a clearer detection was validated by passing the emission source. In practice this means that one sensor node for the right and one for the left side could be more efficient to cover the robot’s surrounding area. By using the front-mounting only one sensor node is required, with the restriction that the detection range is significantly smaller. Thus, it has to be expected that if the robot drops the compounds itself, detection is improbable. An appropriate alternative approach is to mount the sensor node to the back side of the robot. This is, in case of the H20 robot, not feasible since the charging port is at the back side and allows no installation. Depending on the requirements both mounting concepts can be used.

Summarizing, both sensors have shown that they are generally suitable for the detection of VOC leakages in laboratories. While the BME688 reacts comparably slow and shows varying behavior here, for example with acetonitrile the SGP30 sometimes responded comparably low. In combination the sensors gave stable feedback about the gas concentration of the near environment also if the sensors were moved with adequate speeds. Consequently, for optimizing the interpretation of the gas sensor results, a cooperative sensor fusion or artificial-intelligence solutions [35] can be helpful to avoid false negative and false positive errors. Especially, machine learning can help to distinguish different VOCs to precisely identify the real hazard for human operators. The IoT structure supports this approach and enables a central combination of data from the distributed sensor-node network. Further increasing the sampling rate of the sensors can also help to detect VOCs during faster movements of the robot. For both sensors the sampling time of one-second was recommended by the manufacturer and is supported by the sensors pre-calibration as well as the BME library. Using more than two different gas sensors can be helpful, if they are alternatingly triggered. Moreover other sensors, for example, the SGP40 are currently evaluated as candidates for alternative sensor solutions in the presented application.

The application shown is only one example for using the presented sensor node. Further applications for an individual environment-parameter monitoring of transferred samples or human operators are possible. In future works the focus for refining the sensor node is the implementation of the indoor localization to record the position of detection. This enables the addition of the sensor node to all movable elements in the laboratory, for example, roller carriages with laboratory equipment on it, which needs to be monitored and located.

## Figures and Tables

**Figure 1 sensors-21-07347-f001:**
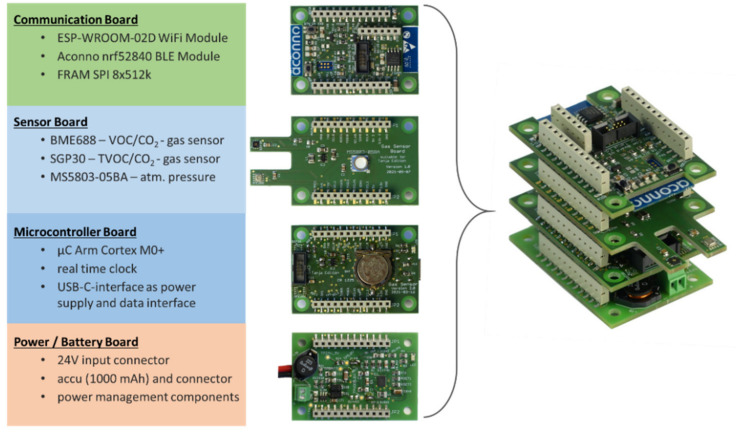
Modules of the sensor node including a brief conclusion of the main components and the assembled sensor-node stack (right).

**Figure 2 sensors-21-07347-f002:**
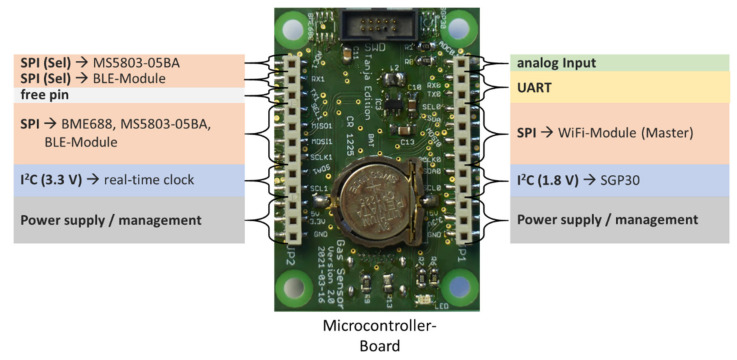
Concluded pin-assignment of the inter-board connection using the microcontroller board as example.

**Figure 3 sensors-21-07347-f003:**
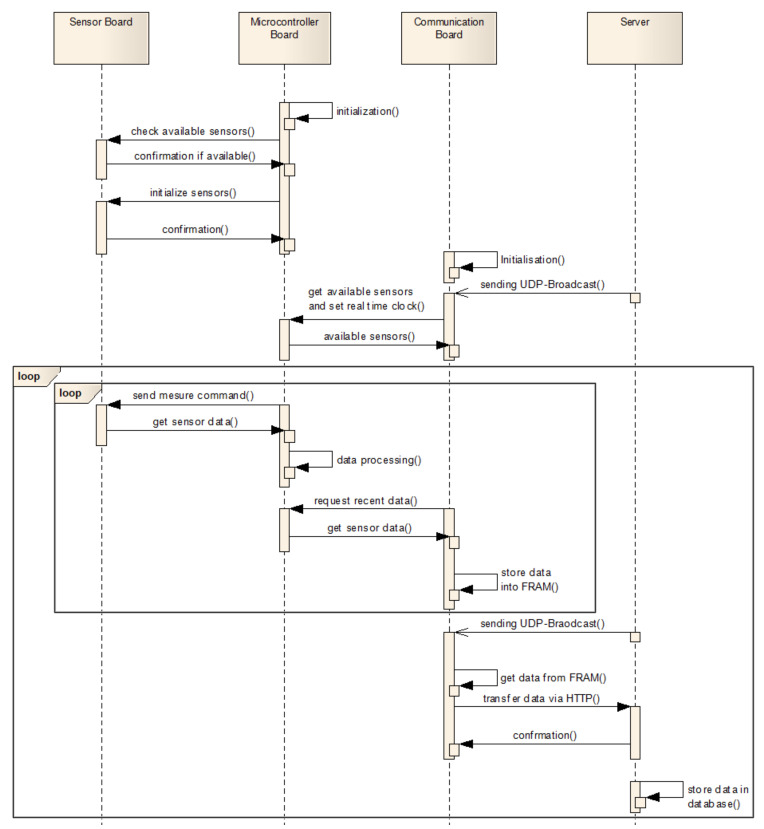
Simplified sequence diagram of the data transfer process between the sensor node components and the communication server.

**Figure 4 sensors-21-07347-f004:**
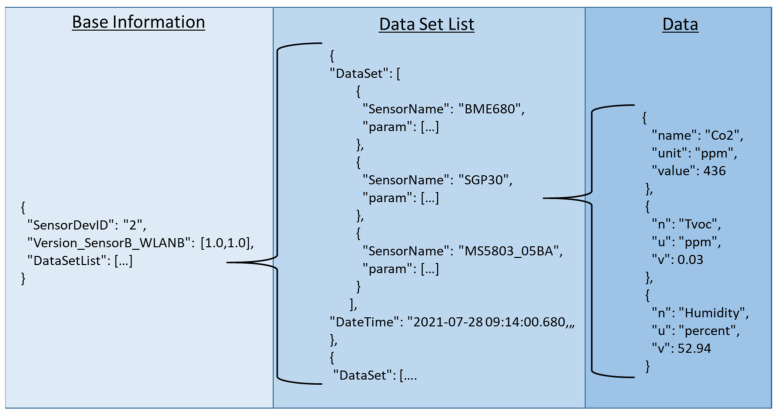
Simplified presentation of the transfer protocol’s structure in JSON-format.

**Figure 5 sensors-21-07347-f005:**
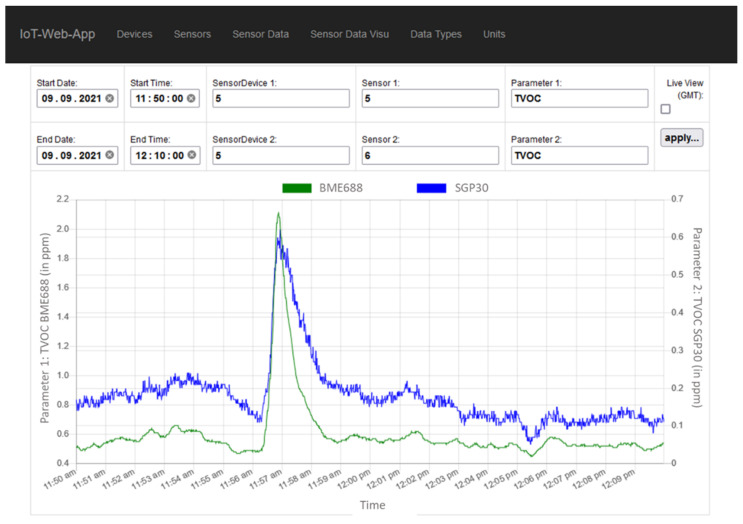
Visualizing tool of the IoT-Web-App showing the responses of the BME688 and the SGP30 to a small VOC emission.

**Figure 6 sensors-21-07347-f006:**
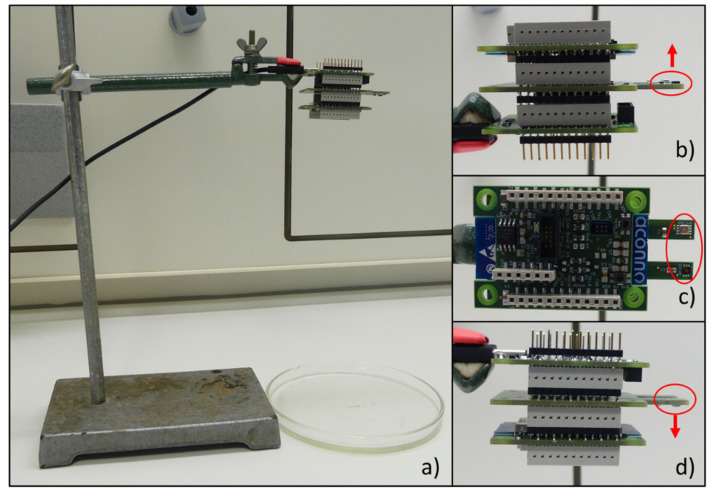
(**a**) Experimental setup in fume cupboard including the stand, petri dish and a sensor node; (**b**) sensor-node orientation: facing upwards; (**c**) sensor-node orientation: facing sideways; (**d**) sensor-node orientation: facing downwards.

**Figure 7 sensors-21-07347-f007:**
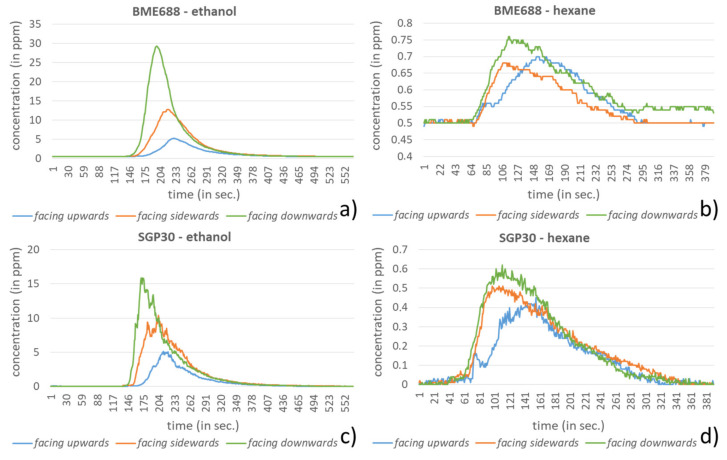
Measurement results for different orientations (facing upwards, sideways and downwards) for the BME688 sensor with (**a**) 10 µL ethanol and with (**b**) 1 mL hexane and the SGP30 sensor with (**c**) 10 µL ethanol and with (**d**) 1 mL hexane.

**Figure 8 sensors-21-07347-f008:**
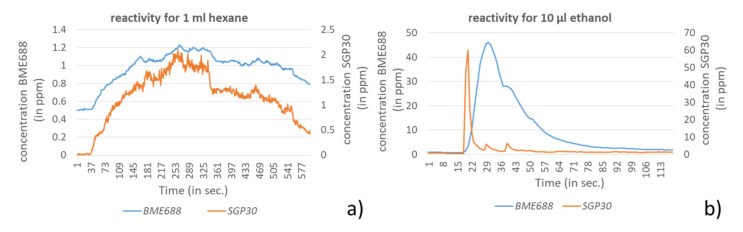
Reactivity of BME688 and SGP30 for (**a**) ethanol and (**b**) hexane.

**Figure 9 sensors-21-07347-f009:**
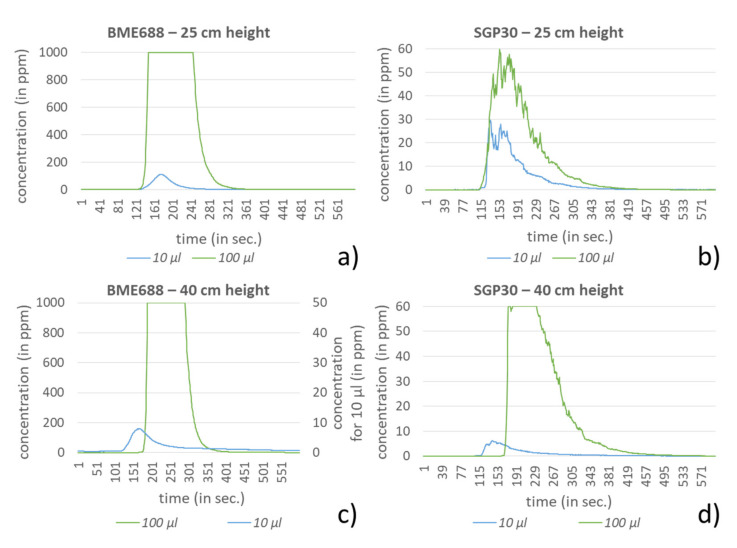
Response of BME688 and SGP30 for different amounts of ethanol and for 25 and 40 cm height above the emission source (consider the second ordinate axis in c) for 10 µL). In (**a**) a measurement height of 25 cm for BME688 and (**b**) for SGP30 is used. In (**c**) the measurement height was changed to 40 cm for BME688 and (**d**) for SGP30.

**Figure 10 sensors-21-07347-f010:**
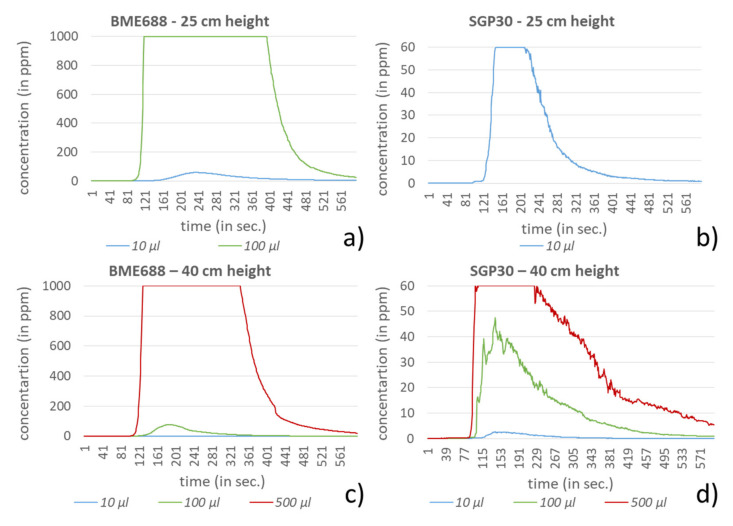
Response of BME688 and SGP30 for different amounts of formic acid and for 25 and 40 cm height above the emission source. In (**a**) a measurement height of 25 cm for BME688 and (**b**) for SGP30 is used. In (**c**) the measurement height was changed to 40 cm for BME688 and (**d**) for SGP30.

**Figure 11 sensors-21-07347-f011:**
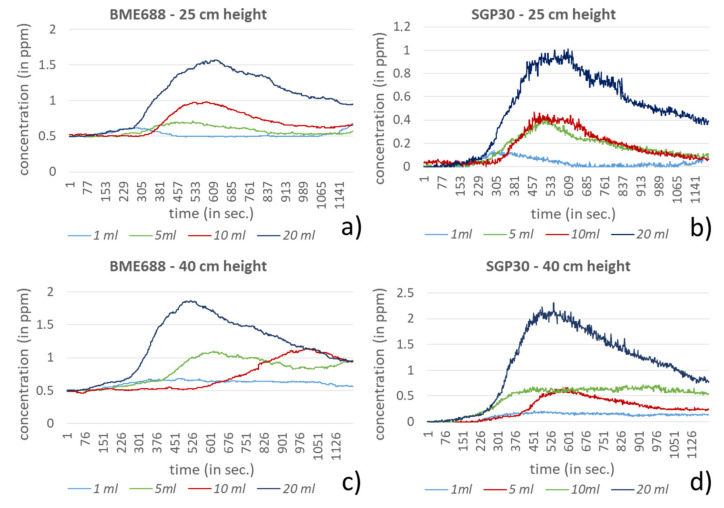
Response of BME688 and SGP30 for different amounts of dichloromethane and for 25 and 40 cm height above the emission source. In (**a**) a measurement height of 25 cm for BME688 and (**b**) for SGP30 is used. In (**c**) the measurement height was changed to 40 cm for BME688 and (**d**) for SGP30.

**Figure 12 sensors-21-07347-f012:**
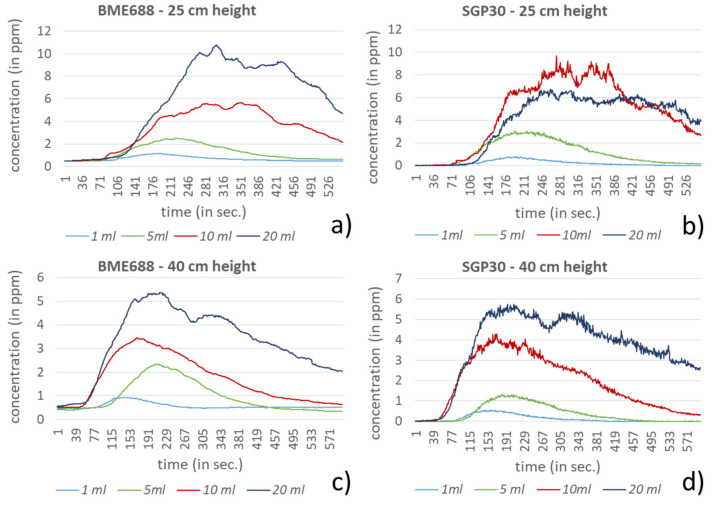
Response of BME688 and SGP30 for different amounts of hexane and for 25 and 40 cm height above the emission source. In (**a**) a measurement height of 25 cm for BME688 and (**b**) for SGP30 is used. In (**c**) the measurement height was changed to 40 cm for BME688 and (**d**) for SGP30.

**Figure 13 sensors-21-07347-f013:**
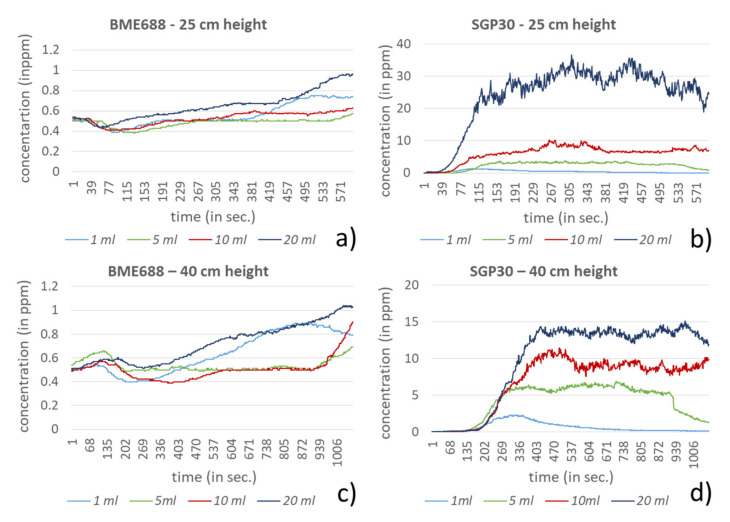
Response of BME688 and SGP30 for different amounts of acetonitrile and for 25 and 40 cm height above the emission source. In (**a**) a measurement height of 25 cm for BME688 and (**b**) for SGP30 is used. In (**c**) the measurement height was changed to 40 cm for BME688 and (**d**) for SGP30.

**Figure 15 sensors-21-07347-f015:**
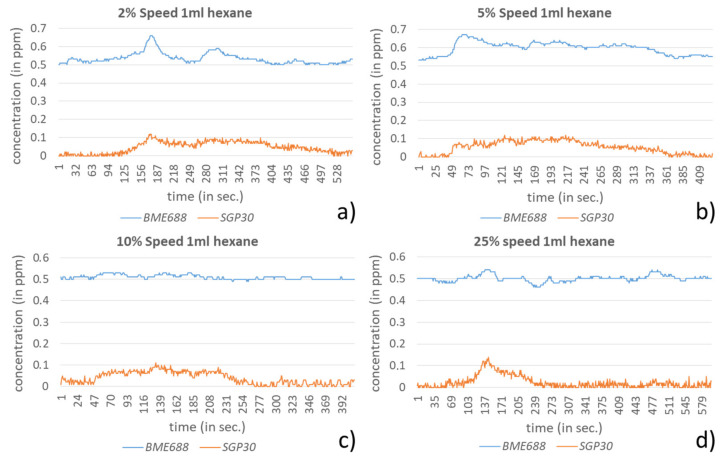
Results for the detection of 1 mL hexane during a pick and place procedure using TS60 in different speed levels, (**a**) for 2%, (**b**) for 5%, (**c**) for 10% and (**d**) for 25%.

**Figure 16 sensors-21-07347-f016:**
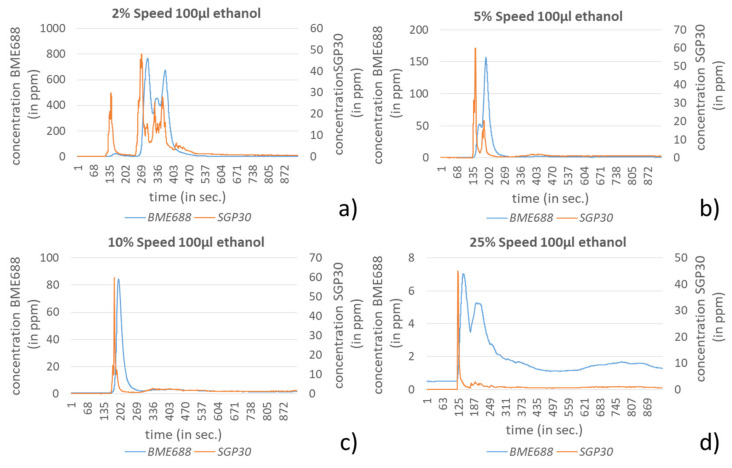
Results for the detection of 100 µL ethanol during a pick and place procedure using a TS60 at different speed levels, (**a**) for 2%, (**b**) for 5%, (**c**) for 10% and (**d**) for 25%.

**Figure 17 sensors-21-07347-f017:**
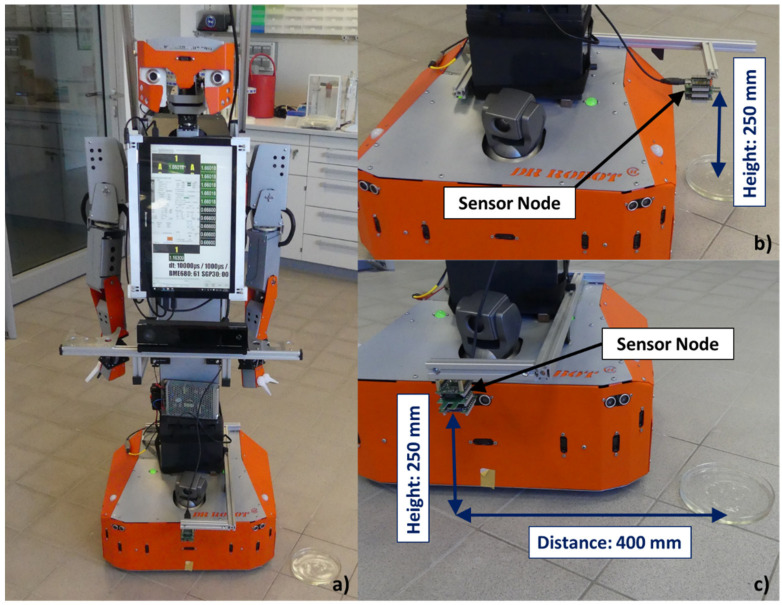
(**a**) H20 robot and the petri dish in a celisca laboratory; (**b**) Experiment setting for the side-mounted sensor node; (**c**) Experiment setting for the front-mounted sensor node.

**Figure 18 sensors-21-07347-f018:**
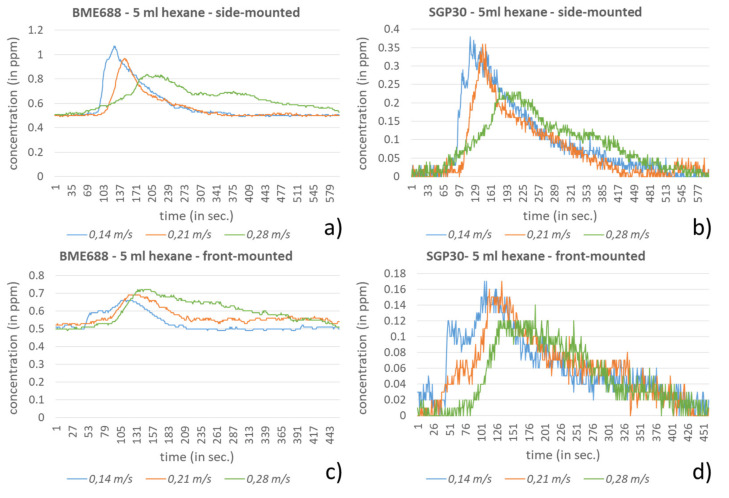
Results for the detection of 5 mL hexane during passing by with the H20-robot in different speed levels and with front- and side-mounted sensor node. (**a**) Response of the side-mounted BME688 and (**b**) SGP30. (**c**) Response of the front-mounted BME688 and (**d**) SGP30.

**Figure 19 sensors-21-07347-f019:**
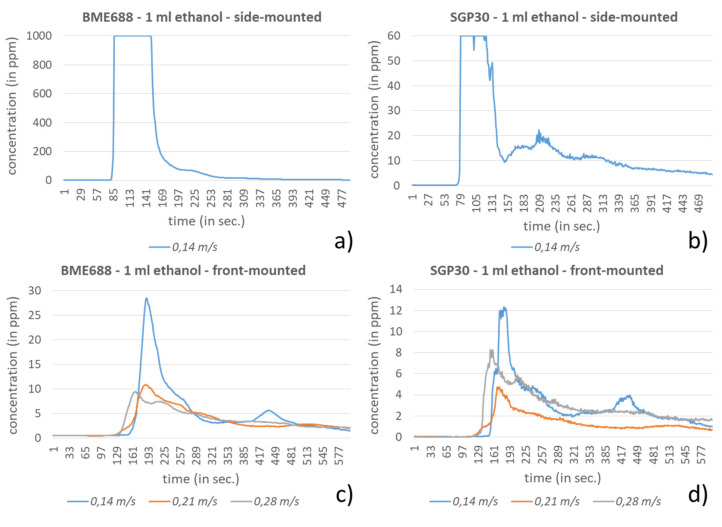
Results for the detection of 1 mL ethanol during passing by with the H20-robot in different speed levels and with front- and side-mounted sensor node. (**a**) Response of the side-mounted BME688 and (**b**) SGP30. (**c**) Response of the front-mounted BME688 and (**d**) SGP30.

**Table 1 sensors-21-07347-t001:** Integrated sensor solutions and their basic specifications.

	BME688 [28]	SGP30 [29]	MS5803-05BA [30]
manufacturer	Bosch	Sensirion	TE connectivity
power supply	1.71–3.6 V	1.62–1.98 V	1.8–3.6 V
acquired parameter (only major parameters included)	IAQ, CO_2_ eq., ambient temperature, relative humidity, atmospheric pressure	TVOC, CO_2_ eq. (H_2_-based)	atmospheric pressure (high resolution), ambient temperature
interfaces	SPI, I^2^C	I^2^C (1.8 V)	SPI, I^2^C
size (in mm³)	3.0 × 3.0 × 0.93	2.45 × 2.45 × 0.9	6.4 × 6.2 × 2.88

## Data Availability

The data presented in this study are available on request from the corresponding author.

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
