# Peer review of "Flexible IoT Gas Sensor Node for Automated Life Science Environments Using Stationary and Mobile Robots"

_sensors, 2021, doi:10.3390/s21217347_

Round 1

Reviewer 1 Report

The manuscript is reporting the evaluation of sensor node suitability as gas detection unit in life science automation structures. In order to do so, the authors used two off the shelf sensors and aimed to investigate the sensor’s behavior towards different VOC gases as well as installation conditions. However, the main research goal is not very clear as the manuscript suffers significantly from lack of novelty, lack of significance and scientific rigor:

  • The abstract of the manuscript is quite vague. It is very general without reporting the main goals achieved by the manuscript.
  • The manuscript is reporting the sensor’s node suitability in life science automation structure and the authors have done several testing experiments in this project. However, the materials used in this project and the methodologies they used to conduct the gas sensing experiments is not explained in the materials and methods section. In fact, no info regarding the materials and the gas sensing test could be find in the methodology section.
  • Furthermore, the testing environment, temperature, pressure, concentration of gases, etc are reported at all.
  • A proper gas sensing chamber with controlled temperature, pressure, gas flow rate and gas concentration is necessary for any gas sensing research. Here, however, the authors simply used a microlitre volume of liquid in a petri dish and conducted the gas sensing test. How such experimental design could ensure reliability in collected data?
  • The quality of the graphs presented in this manuscript is quite poor. For example, the data (and text) presented in Figure 5 is not even readable. The quality of the graphs presented in Figures 7-13, Figures 15-16 and Figures 17-19 is below the standard/acceptable level for scientific publication.
  • The graphs presented in Figures 9, 10 and 19a-b are quite confusing. Why are the top parts of the graphs cut? Is this due to sensor’s saturation? If yes, a lower gas concentration must be used for sensing evaluation, as the data presented here are not scientifically acceptable!
  • Similar to the abstract, the conclusion section of this manuscript is very vague. In fact, nothing significant was reported in the conclusion as the scientific summary of this project.

Author Response

Dear Sir or Madam,

thank you very much for your remarks. We adapted our paper accordingly and replied to your remarks in the attached file.

Kind regards

Sebastian Neubert

Reviewer 2 Report

Through my evaluation, I found that the paper fits into the scope of the journal very well. I am convinced by most parts of the results. I recommend the authors address the following comments, before the acceptance of the paper.

  1. It is well established in the literature that MOX type sensors suffer from crosstalk, deviation in temperature and humidity. This may cause a false gas sensor response with an interfering gas analyte. The author does not discuss in the paper or show data:
    1. If the sensor can maintain the sensor response in a dynamic humidity environment as outlined in line 199.
    2. Or if the experiment was conducted in a control temperature environment.
    3. Explore the crosstalk with other known organic compounds or gas analytes that are known in the literature to affect MOx (e.g., Acetone, Isopropanol, NO2, CO2, H2, CH4).
  2. It would be good if the authors show results gas sensing response and recovery to a reference baseline e.g., in the air without exposure to gas/vapour analytes.
  3. As mentioned in question 1, the author shows ethanol and hexane sensor response, and it seems that ppm response is relative to baseline. If this is the case what happens if ethanol and hexane vapours are mixed? How does the sensor distinguish the crosstalk?
  4. The crosstalk and interfering gases or vapours that be potentially solved by cloud-based AI-machine learning models described in this paper. Some AI-driven sensors literature, including Advanced Intelligent Systems, 2020, 2, 2000063, should be cited.

Author Response

(The authors gave the same response as above.)

Round 2

Reviewer 1 Report

The authors have addressed all the issues raised by the Reviewer. The manuscript is now acceptable for publication. 

Reviewer 2 Report

I am satisfied with the revised version.